# Indian Ocean Dipole leads to Atlantic Niño

Lei Zhang [1] ✉ & Weiqing Han[1]

Atlantic Niño is the Atlantic equivalent of El Niño-Southern Oscillation (ENSO), and it has prominent impacts on regional and global climate. Existing studies suggest that the Atlantic Niño may arise from local atmosphere-ocean interaction and is sometimes triggered by the Atlantic Meridional Mode (AMM), with overall weak ENSO contribution. By analyzing observational datasets and performing numerical model experiments, here we show that the Atlantic Niño can be induced by the Indian Ocean Dipole (IOD). We find that the enhanced rainfall in the western tropical Indian Ocean during positive IOD weakens the easterly trade winds over the tropical Atlantic, causing warm anomalies in the central and eastern equatorial Atlantic basin and therefore triggering the Atlantic Niño. Our finding suggests that the cross-basin impact from the tropical Indian Ocean plays a more important role in affecting interannual climate variability than previously thought.

[1] Department of Atmospheric and Oceanic Sciences, University of Colorado, Boulder, CO, USA. ✉email: lezh8230@colorado.edu

In the tropical Pacific and Atlantic Oceans, the mean sea surface temperature (SST) is warm in the west and cold in the east, under the influence of prevailing easterly trade wind forcing. The east-west SST gradient, however, is substantially weakened or even reversed in some years, which is accompanied by a significant slackening of the trade winds[1]. This climate phenomenon is called El Niño, the positive phase of El Niño-Southern Oscillation (ENSO), in the tropical Pacific[2] and is referred to as the Atlantic Niño in the tropical Atlantic[3].

The Atlantic Niño is associated with prominent climate anomalies across the tropical Atlantic and has far-reaching impacts on climate in remote regions[3–6]. For instance, the Atlantic Niño enhances seasonal rainfall over South America and Africa[7,8]; it can also affect the African Easterly Jet[9], which can subsequently influence Atlantic hurricanes[10,11]. It has been suggested that the Atlantic Niño may strengthen the Pacific Walker circulation and lead to La Niña[12–15], the negative phase of ENSO. Through atmospheric teleconnection, the Atlantic Niño can also weaken the Indian summer monsoon[16–18]. Therefore, understanding and predicting the Atlantic Niño are of paramount importance and have large societal benefits.

Unlike ENSO that peaks in boreal winter, the Atlantic Niño is generally phase-locked to boreal summer[19,20], but some events may peak in late fall and winter[21,22]. Its development involves positive feedbacks between atmosphere and ocean[3,22–24]. During the Atlantic Niño, warm SST anomalies (SSTA) appear in the central and eastern tropical Atlantic basin, which lowers the sea level pressure (SLP) in the region. The associated eastward SLP gradient force induces westerly wind anomalies, which cause anomalous surface Ekman convergence, deepen the thermocline, reduce oceanic upwelling cooling and thus further increase the warm SSTA in the eastern basin. In addition, the anomalous westerlies weaken the westward South Equatorial Current and produce eastward heat advection anomalies, enhancing the warm SSTA. Surface heat flux anomalies do not contribute much to the warm SSTA associated with the Atlantic Niño[25].

Some existing studies suggest that remote forcing from the Pacific El Niño and the Atlantic Meridional Mode (AMM)[26] may help trigger the Atlantic Niño. El Niño can induce wind anomalies over tropical Atlantic[27,28], which excite oceanic waves that warm the eastern tropical Atlantic SST[20]. This remote ENSO influence however is overall weak[29], partly due to the opposing effects of the anomalous equatorial easterlies (which generate upwelling Kelvin wave) and off-equatorial anticyclonic wind anomalies (which generate downwelling Rossby wave)[30]. Indeed, warm but weak SSTA is only found north of the Atlantic equator during El Niño (Supplementary Fig. 1)[29,30]. On the other hand, surface wind anomalies associated with the AMM have been shown to cause the Atlantic Niño through exciting oceanic Kelvin and Rossby waves and/or anomalous temperature advection[24,25,31]. A recent observational study, however, shows that many Atlantic Niño events occur without being preconditioned by the AMM[21]. While the development of Atlantic Niño is typically associated with weakened Atlantic trade winds[5,6], it remains unclear what may have caused the wind anomalies.

It has been recognized that inter-basin interactions among the three tropical ocean basins play a crucial role in shaping climate variability and long-term climate change[32,33]. Whilst most existing studies focus on tropical Indian–Pacific and Pacific–Atlantic interactions, interaction between the tropical Indian and Atlantic Oceans is much less explored. The role played by variability of the tropical Indian Ocean in triggering the Atlantic Niño has not been investigated thus far. In the tropical Indian Ocean, the dominant interannual climate variability mode that can operate independently from ENSO is the Indian Ocean

Dipole (IOD)[34,35], which is manifested as an east-west dipole-like SSTA pattern, begins to develop in boreal summer and peaks in boreal fall.

Here, we show observational evidence that positive phase of the IOD (pIOD) can cause westerly wind anomalies over the tropical Atlantic Ocean, weakening the easterly trade winds and causing warm SSTA. Consequently, the pIOD can trigger Atlantic Niño through atmospheric teleconnection. We confirm the associated physical mechanisms by performing various model experiments using standalone atmospheric and oceanic models, and fully coupled climate models.

## Results

**Impact of IOD on Atlantic Niño in observations.** To understand the possible impact of the IOD on Atlantic Niño, we first perform a lead-lag regression of SST and wind anomalies on the Dipole Mode Index (DMI), an index for the IOD defined as the difference of SSTA between western and eastern tropical Indian Ocean[34]. Similar SSTA regression patterns are obtained when the ENSO influence is removed (compare Fig. 1 and Supplementary Fig. 2), which induces basin-wide tropical Indian Ocean warm SSTA[36], but weak SSTA over the Atlantic Niño region[37,38] (Supplementary Fig. 1). Our results show that SSTA of the pIOD starts to develop ~5 months prior to its peak, accompanied by easterly wind anomalies over the tropical Indian Ocean (Figs. 1b, c and 2). As the pIOD further develops and reaches its peak, both the dipole-like SSTA and the associated easterly wind anomalies in the tropical Indian Ocean strengthen (Fig. 1d, e), and prominent positive (negative) rainfall anomalies are found in the western (eastern) tropical Indian Ocean (Supplementary Fig. 3d, e). In the meantime, westerly wind anomalies start to strengthen over the tropical Atlantic, along with northerly anomalies off the west coast of South Africa, weak easterly near ~15°S, and southerly over eastern South America (Fig. 1e). These large-scale wind anomalies form a lower-level cyclonic (clockwise) circulation over the south tropical Atlantic Ocean, which is part of atmospheric Rossby wave structure as a Gill-type response to the positive rainfall anomalies over the western pole of the pIOD[39]. During the pIOD decaying phase, the SST, wind, and rainfall anomalies over the tropical Indian Ocean weaken, while the tropical south Atlantic clockwise wind anomalies maintain and enhanced warm SSTA is seen in the central and eastern Atlantic basin (Figs. 1f, 2 and Supplementary Fig. 3f). Meanwhile, rainfall is enhanced over the equatorial Atlantic Ocean and suppressed to the north, suggesting the equatorward shift of the Inter Tropical Convergence Zone (ITCZ)—a feature that is typically observed during the Atlantic Niño.

These observational analyses suggest that the Atlantic warm SSTA peaks in 3–5 months after the pIOD peak (Fig. 2), when the SST and wind anomalies associated with the pIOD in the Indian Ocean decay significantly (Figs. 1f, g and 2). Similarly, composite of the Atlantic Niño events that are associated with pIOD forcing shows that the Atlantic Niño signals start to appear prior to the pIOD peak, which leads the Atlantic Niño peak by 2–4 months (Supplementary Fig. 4). Given that the evolutions of both IOD and Atlantic Niño exhibit strong seasonal dependence, we further performed regression of 3-month mean SSTA on the September–November (SON)-mean DMI, which also shows a lead of the pIOD peak over the Atlantic Niño peak by 2–4 months (Supplementary Fig. 5). These results suggest that pIOD can induce Atlantic Niño, with the latter lagging by 1–5 months depending on different cases (Supplementary Fig. 6). The lead time spread is because while pIOD may trigger Atlantic Niño and enhance its initial development through atmospheric

## Lead-lag regression of SST and 850hPa winds on DMI

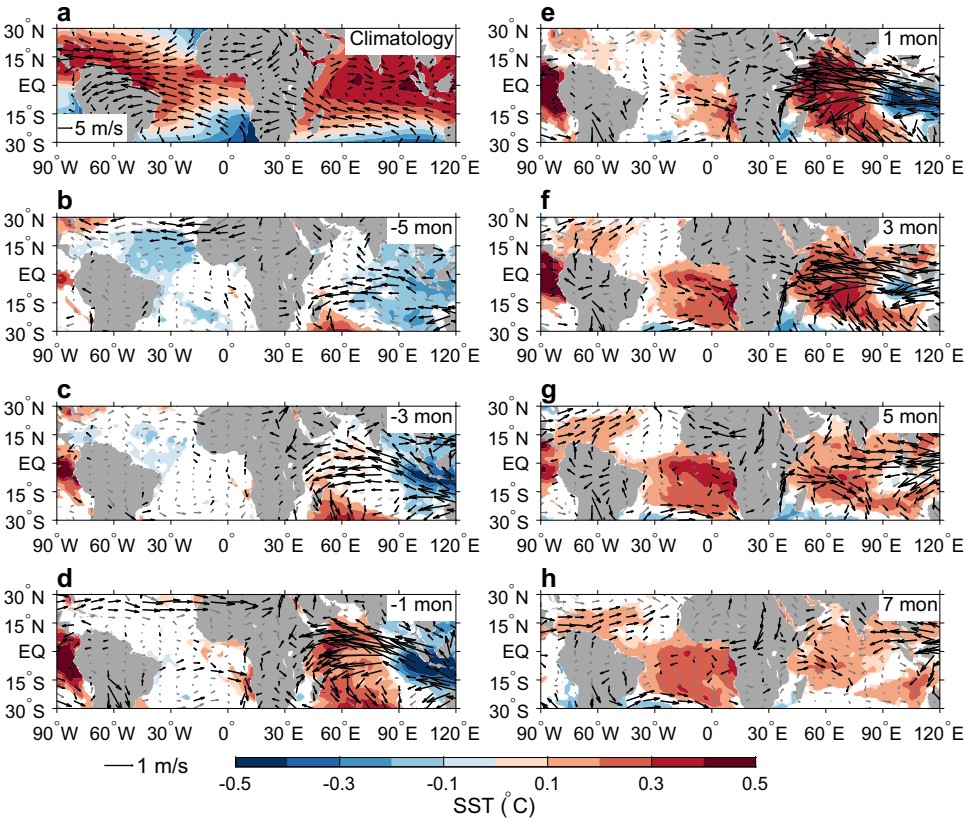

**Fig. 1 Impact of positive IOD on Atlantic Niño. a** Annual mean climatology of sea surface temperature (SST) (°C) and 850hPa wind (m s⁻¹). **b–h** Lead-lag regression of monthly SST anomalies (SSTA; shading; °C °C⁻¹) from the Hadley Centre Sea Ice and SST (HadISST) and 850 hPa wind anomalies (vector; m s⁻¹ °C⁻¹) from the European Centre for Medium-Range Weather Forecasts (ECMWF) twentieth-century reanalysis (ERA-20C) on monthly Dipole Mode Index (DMI), which represents the Indian Ocean Dipole (IOD) mode. **b–h** Results for −5 to 7 months leading time, with negative representing months prior to IOD peak and positive for after IOD peak. Shading and black vectors represent anomalies that are statistically significant at the 90% confidence level.

teleconnection, the growth of Atlantic Niño to its mature phase depends on local large-scale air-sea interaction, which can be affected by both local and remote forcing (such as extratropical processes).

**Relevant processes for Atlantic Niño warm anomalies**. The generation of the Atlantic Niño by the pIOD is initiated by the westerly wind anomalies associated with atmospheric Rossby wave's lower-level cyclones over the tropical Atlantic Ocean. The anomalous equatorial westerlies induce eastward oceanic currents (Figs. 1c, d and 3c, d), advecting the warm water from the warm pool in the west (Fig. 1a) to the central and eastern equatorial Atlantic basin (Fig. 3). In addition, the anomalous equatorial westerlies also cause anomalous equatorial Ekman convergences, which increase sea level (Fig. 3e, f), deepen the thermocline, and suppress the mean oceanic upwelling induced by the easterly trades (Figs. 1a and 3a). Analysis of satellite-derived daily sea level data during 1993–2019 also shows that downwelling equatorial Kelvin waves induced by the westerly wind anomalies propagate from the central to the eastern equatorial Atlantic Ocean in some cases (Supplementary Fig. 7), which further contribute to the warm SST in the region. Similar eastward propagation of easterly wind anomalies-induced upwelling Kelvin waves is found during Atlantic Niña. The two processes together are the primary causes for the warm SSTA in the eastern Atlantic basin. Surface heat flux

anomalies do not contribute to the warm SSTA of the Atlantic Niño (Supplementary Fig. 8), which is in line with the previous studies[25].

We note that there are prominent warm SSTA off the west coast of Angola that first appear during the pIOD developing phase and sustain through the Atlantic Niño peak (Figs. 1 and 2). This enhanced coastal warming is likely caused by the local northerly wind anomalies along the coast (part of the atmospheric Rossby wave response to the pIOD) that weaken the Benguela current—a northward cold current—and suppress the coastal upwelling through inducing anomalous surface Ekman convergence to the coast (Figs. 1 and 3)[40,41]. In addition, the equatorial westerly wind anomalies may also contribute to the enhanced coastal warm SSTA through the downwelling equatorial Kelvin waves discussed above and the subsequent poleward-propagating coastally-trapped waves along the eastern boundary (Supplementary Fig. 7), as suggested in the previous studies[42,43]. The coastal SST warming to the west of Angola has been previously referred to as the Benguela Niño[44]. The Atlantic Niño and the Benguela Niño indeed tend to co-occur, and the two are sometimes viewed as one phenomenon[45]. It has been suggested that the Benguela Niño leads the Atlantic Niño by ~1 month due to the shallower thermocline along the Benguela coast, which makes the SST change more sensitive to upwelling anomaly[45]. This is consistent with the earlier appearance of the Benguela Niño than the Atlantic Niño in Fig. 1.

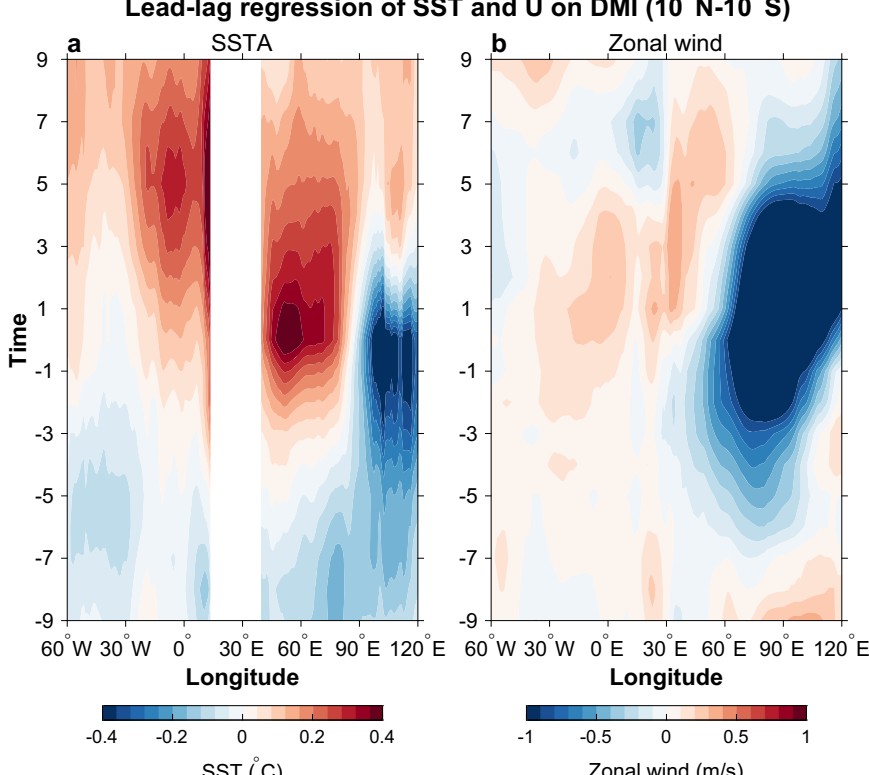

**Fig. 2 Hovmöller diagrams showing evolutions of sea surface temperature (SST) and zonal wind anomalies associated with positive IOD.** Lead-lag regression of (**a**) SST (°C °C$^{-1}$) and (**b**) 850 hPa zonal wind anomalies (m s$^{-1}$ °C$^{-1}$) on the Dipole Model Index (DMI) averaged between 10°N–10°S. Y-axis represents the leading time, with negative representing months prior to Indian Ocean Dipole (IOD) peak and positive for after IOD peak.

**Numerical model experiments**. To provide further evidence for our proposed mechanism that the pIOD can induce westerly wind anomalies over the tropical Atlantic Ocean, we perform atmospheric general circulation model (AGCM) experiments forced with the tropical Indian Ocean SSTA associated with the IOD. The AGCM results show that the pIOD indeed induces prominent westerly wind anomalies over the equatorial Atlantic Ocean during its developing and peak phases (Fig. 4a, b). Experiment using a linear continuously stratified ocean model forced by these westerly wind anomalies produces lower (higher) sea level in the western (eastern) equatorial Atlantic Ocean, as well as prominent eastward ocean current anomalies (Supplementary Fig. 9), consistent with the observational results (Fig. 3e). These results suggest that the pIOD indeed can contribute to the formation of the Atlantic Niño. During the pIOD decaying phase, the tropical Atlantic wind response in the AGCM becomes weaker because the warm SSTA pole of the pIOD is shifted to the central tropical Indian Ocean south of the equator, further away from the tropical Atlantic Ocean. Hence, the observed tropical Atlantic westerly wind anomalies after the pIOD peak (Fig. 1f) are likely due to the local atmosphere-ocean interaction, after the pIOD initiated the warm SSTA. We also note that the observed northerly wind anomalies off the west coast of Angola are not captured by the AGCM, which could be due to the model's low resolution and bias in simulating such coastal wind anomalies.

Given that large-scale atmosphere-ocean interaction, i.e., the Bjerknes feedback[1], plays a crucial role in the growth of the Atlantic Niño to its mature stage, we further performed pacemaker experiments using a coupled climate model, in which the Indian Ocean SSTA is nudged to the observed values whereas atmosphere and ocean are fully coupled elsewhere (see "Methods" section; Supplementary Fig. 10). The pacemaker experiment,

therefore, isolates the effect of the Indian Ocean SSTA on Atlantic Niño, while fully representing the role of air-sea interaction. Like the observational results, pacemaker experiment shows that as the pIOD develops (Supplementary Fig. 11), rainfall over the western tropical Indian Ocean enhances, which drives westerly wind anomalies over the tropical Atlantic Ocean that trigger the Atlantic Niño, and contributes to its development even prior to the pIOD peak (Fig. 5). As the pIOD decays, the Atlantic Niño continues developing through local air-sea interaction. Consequently, the peak of the central-eastern equatorial Atlantic warming lags that of the pIOD by ~3 months in the model, which agrees with observational results. Hence, the Indian Ocean pacemaker experiment further supports the proposed atmospheric teleconnection through which the pIOD can trigger the Atlantic Niño and affect its development.

## Discussion

Interbasin interaction plays an important role in global climate variability and change, and have attracted considerable attention recently[32,33]. In particular, it has been recognized that despite the relatively small basin size, the tropical Indian Ocean plays an increasingly important role in influencing climate conditions in both Pacific and Atlantic Oceans from interannual to centennial timescales[46–52]. This is in part because of the basin-wide high background SST in the tropical Indian Ocean that is close to the threshold for tropical deep convections[53,54], which allows a small change in SST to drive unproportionally large atmospheric response.

Here, by carrying out observational analysis and performing numerical model experiments, we show that the IOD—the dominant interannual climate variability mode in the tropical

## Sea level and upper ocean currents

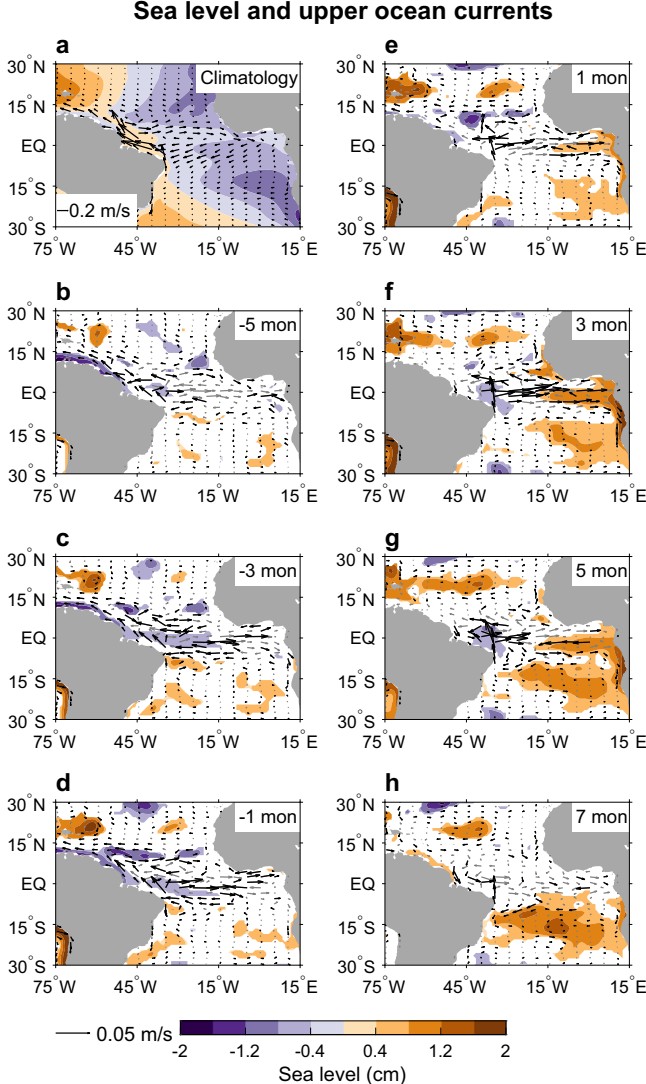

**Fig. 3 Sea level and near surface current in the tropical Atlantic Ocean associated with positive IOD. a** Annual mean climatology of sea level (cm) and ocean currents averaged over the upper 50 m (m s$^{-1}$). **b–h** Lead-lag regression of sea level (shading; cm $^\circ$C$^{-1}$) and ocean current anomalies averaged over the upper 50 m (vector; m s$^{-1}$ $^\circ$C$^{-1}$) from the European Centre for Medium-Range Weather Forecasts (ECMWF) Ocean Reanalysis System 4 (ORAS4) on monthly Dipole Mode Index (DMI). **b–h** Results for —5 to 7 months leading time, with negative representing months prior to Indian Ocean Dipole (IOD) peak and positive for after IOD peak. Shading and black vectors represent anomalies that are statistically significant at the 90% confidence level.

Indian Ocean that can operate independently from ENSO—can trigger the Atlantic Niño through the atmospheric bridge. Our results show that during the positive IOD, the above-normal rainfall over the western tropical Indian Ocean causes westerly wind anomalies over the tropical Atlantic Ocean, weakening the tropical Atlantic easterly trade winds. The westerly wind anomalies excite oceanic downwelling Kelvin waves, cause anomalous eastward ocean current, and suppress oceanic upwelling, resulting in warm anomalies in the equatorial Atlantic Ocean. After the IOD peak, local atmosphere-ocean interaction kicks in and leads to further development of the Atlantic Niño. As a result, the peak of Atlantic Niño lags that of the IOD by ~3–5 months.

Both observations and model experiments suggest that ~1/3 of Atlantic Niño events are contributed by the remote IOD forcing. In observations, out of total 31 (30) Atlantic Niño (Niña) events from 1950 to 2019, 10 (10) were preceded by positive (negative) IOD events, and the composited SSTA evolution resembles that of the regression pattern on the DMI (Fig. 1 and Supplementary Fig. 4). Similarly, the ensemble mean results from pacemaker experiment show that 32–34% of Atlantic Niño and Niña events are associated with the IOD forcing, and their amplitudes are slightly higher than the events that are independent of the IOD (Supplementary Fig. 12). To increase the sample size, we also analyzed the 40-member ensemble simulations from the Community Earth System Model (CESM)[55], and also found that ~24–25% of Atlantic Niño and Niña were led by IOD prior to their peaks.

In addition to the triggering effect, the pIOD forcing also seems to affect timing of the Atlantic Niño. Compared to the independent Atlantic Niño that tends to peak in boreal summer, most of the Atlantic Niño events that peak during boreal fall and winter especially from December to January, which have been referred to as Atlantic Niño II[22] or late-onset Atlantic Niño[21], are associated with the pIOD forcing (Supplementary Fig. 13a). Similarly, the Atlantic Niña events associated with the negative IOD (nIOD) also tend to peak during October–February, with none of them occurring between July and September (Supplementary Fig. 13b). Interestingly, a positive IOD occurred in 2019[56], which is the strongest event since 1979 when the satellite observations become available, and it is followed by an Atlantic Niño event whose amplitude is also the strongest in the past few decades[21]. Indeed, Indian Ocean pacemaker experiments show that the positive IOD that peaks in October 2019 leads to a strong Atlantic Niño event that peaks in December (Supplementary Fig. 14).

The reason that the impact of IOD on Atlantic Niño is more prominent during boreal fall and winter seasons is related to its seasonality, since the IOD usually develops in the summer and peaks in the fall. In addition, the relatively low mean state SST in the western tropical Indian Ocean during boreal summer under the influence of the Indian summer monsoon compared to the higher mean SST during fall and winter could be another reason, because a higher background SST allows the tropical Indian Ocean SSTA to affect local rainfall more effectively[53,54] and thus have a stronger impact on the tropical Atlantic Ocean through the atmospheric teleconnection.

Given the prominent impact of the IOD on the Atlantic Niño, our finding provides an additional source of predictability for the Atlantic Niño. Additionally, given the prominent role of the Atlantic Niño in affecting the ENSO development[12-15], our results point to a potential mechanism for IOD to affect ENSO with Atlantic Niño as a bridge. Therefore, it is important to assess whether current climate forecast models can faithfully represent the IOD-Atlantic Niño relationship, successful simulation of which could help improve seasonal climate prediction.

## Methods

**Observational data sets.** In order to examine large-scale climate anomalies associated with tropical climate modes, we analyze SST data during 1950–2019 from the Hadley Centre Sea Ice and SST (HadISST)[57], 850 hPa wind and precipitation data during 1950–2010 from the European Centre for Medium-Range Weather Forecasts (ECMWF) twentieth-century reanalysis (ERA-20C)[58], and sea level data during 1958–2017 from ECMWF Ocean Reanalysis System 4 (ORAS4)[59]. For comparison, we also analyze satellite-derived daily sea level data during 1993–2019 obtained from Copernicus Marine Environment Monitoring Service (CMEMS), and we remove the first three harmonics of daily climatology to obtain daily anomalies. To analyze the physical causes for the SSTA, we use surface heat flux data from ERA-20C.

To document climate variable anomalies associated with tropical climate modes, we calculate climate indices using SST data. For ENSO, we use the Niño-3.4 index defined as SSTA averaged over (170°W–120°W; 5°S–5°N). For the IOD, we

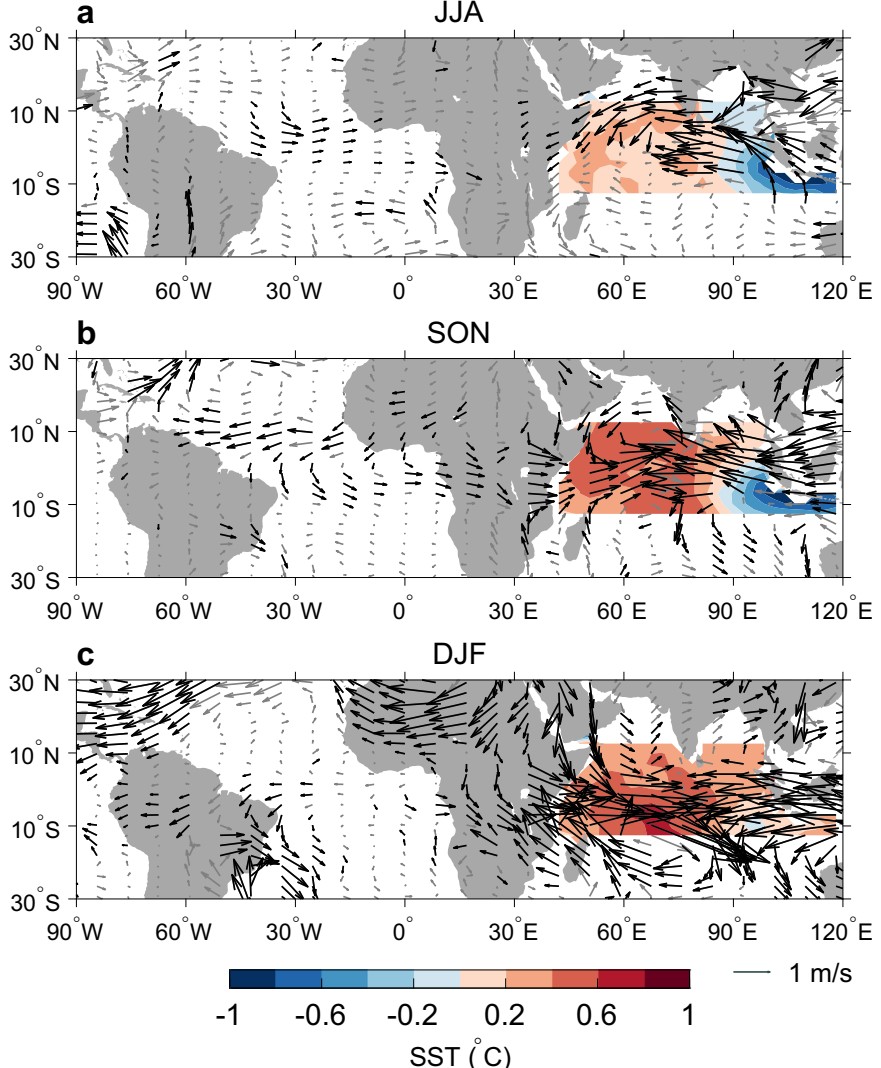

**Fig. 4 Atmospheric model results forced with IOD sea surface temperature anomalies (SSTA).** 850 hPa wind anomalies (vector; m s$^{-1}$) caused by SSTA associated with the Indian Ocean Dipole (IOD) in the atmospheric general circulation model (AGCM) experiments. **a–c** Anomalies averaged during June–August (JJA), September–November (SON), and December–February (DJF), respectively. Shown are differences between AGCM experiments forced by positive and negative IOD. Black vectors denote the wind differences that are statistically significant at the 90% confidence level.

use the Dipole Mode Index (DMI)[34], defined as differences of SSTA averaged over the western (50°E–70°E; 10°S–10 °N) and eastern (90°E–110°E; 10°S–0°) tropical Indian Ocean. For the Atlantic Niño, we use the SSTA averaged over the ATL3 region (20°W–0°; 3°S–3 °N). In this study, we define positive IOD (pIOD) events as when the peak of 3-month smoothed DMI exceed its standard deviation, and negative IOD (nIOD) events as when the smoothed DMI falls below negative standard deviation. Similarly, Atlantic Niño events are defined as when the peak of the 3-month smoothed ATL3 index exceeds the one standard deviation threshold. Based on these criteria, we have selected 31 Atlantic Niño and 30 Atlantic Niña events during 1950–2019. Furthermore, if an observed Atlantic Niño is preceded by a pIOD within 5 months prior to its peak, the Atlantic Niño is considered associated with pIOD forcing, and similarly for observed Atlantic Niña and nIOD. We have selected 10 such Atlantic Niño events (32%), and 10 Atlantic Niña events associated with nIOD (33%) in observations (Supplementary Figs. 4, 6, and 13). The reason that we did not use seasonal mean climate mode indices to define the climate events—for instance, using June–August (JJA)-mean ATL3 index to define the Atlantic Niño/Niña—is because although the Atlantic Niño events generally tend to peak in boreal summer, many of them occur in other seasons, including the 2019 super Atlantic Niño event that peaks in December 2019–January 2020. The only exception is for the construction of SSTA forcing for the atmospheric model (see below), for which monthly SSTA associated with the IOD is added to monthly SST climatology. Hence, seasonality needs to be considered, and we defined and selected the positive and negative IOD years as when the SON-mean DMI exceeds or falls below one standard deviation.

To remove the influences of anthropogenic global warming, we remove the linear trend in all monthly climate variable anomalies. To exclude the ENSO-related signals (e.g., Supplementary Fig. 2), we remove the 3-month lead-lag regression on the Niño-3.4 index from the anomalies. To remove the influences of subseasonal variability, we use 3-month running mean SSTA when performing the composite analysis (e.g., Supplementary Fig. 4).

**CESM Large ensemble**. In order to analyze the IOD-Atlantic Niño relationship with a larger sample size than using the observational data, we analyzed monthly SST data from the historical simulations (1950–2005) and future projections (2006–2019) from the forty-member ensemble of National Center for Atmospheric Research (NCAR) Community Earth System Model version 1 (CESM1) large ensemble (CESM1-LE)[55] for comparison. Prior to analysis, we removed ensemble mean results from each ensemble member to remove the externally forced signals.

**Model experiments**. To further confirm our proposed mechanism that the IOD affects the tropical Atlantic wind anomalies through the atmospheric bridge, we perform atmospheric general circulation model (AGCM) experiments using ECHAM4.6[60] from Max Planck Institute (MPI) in Hamburg (HAM). Model horizontal resolution is ~2.8°, with 19 vertical levels. We performed two sets of AGCM experiments, forced by positive and negative IOD SSTA in the tropical Indian Ocean, respectively. Outside the tropical Indian Ocean, SST is set to monthly climatology. The SSTA forcing was obtained by computing the composite

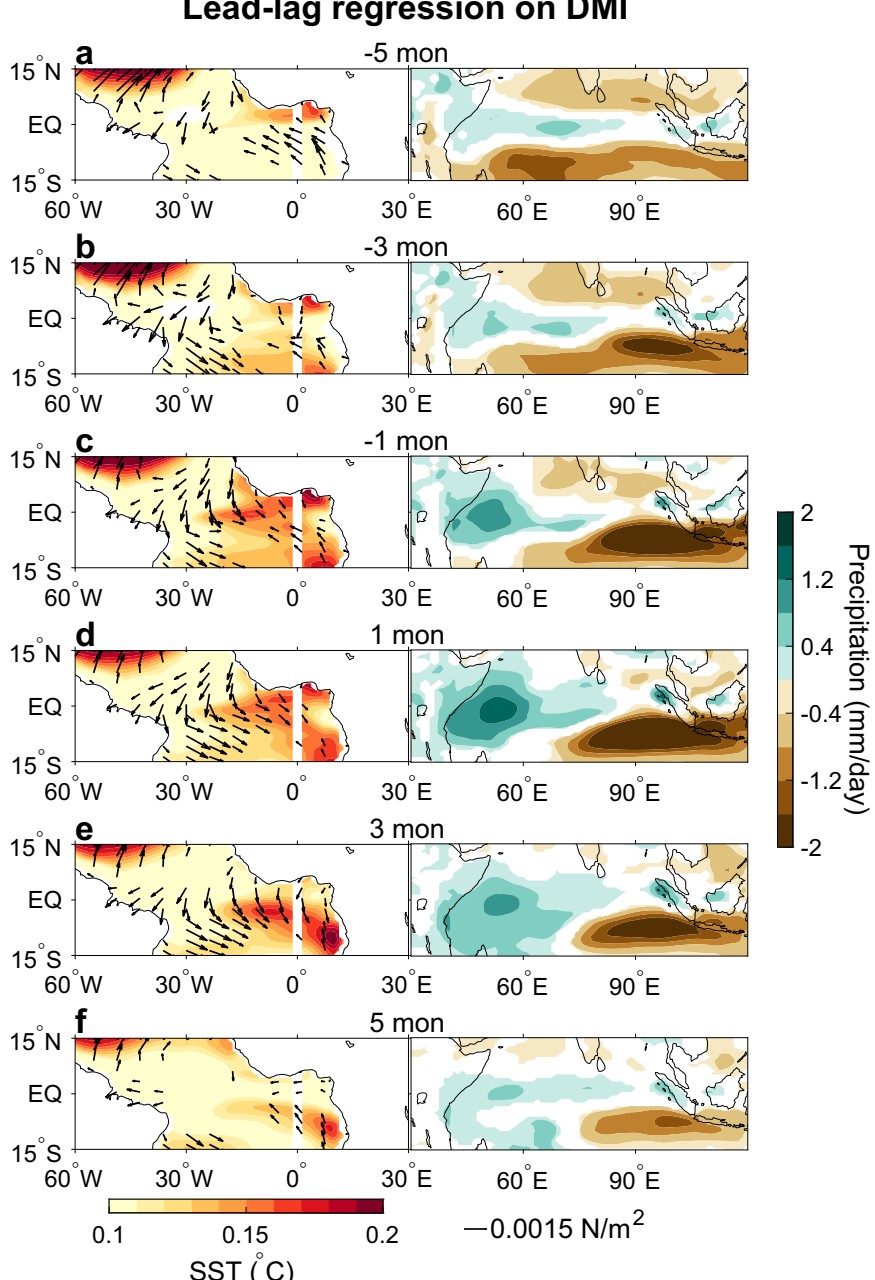

**Fig. 5 Coupled climate model experiment assessing Indian Ocean sea surface temperature (SST) effect.** Lead-lag regression of SST (shading; °C °C$^{-1}$) and surface wind stress (N m$^{-2}$ °C$^{-1}$) anomalies in the tropical Atlantic Ocean and precipitation anomalies (shading; mm day$^{-1}$ °C$^{-1}$) in the tropical Indian Ocean on monthly Dipole Mode Index (DMI). Shown are ensemble mean Indian Ocean Pacemaker experiment results that are statistically significant at the 90% confidence level. **a–f** Results for —5 to 5 months leading time, with negative representing months prior to Indian Ocean Dipole (IOD) peak and positive for after IOD peak.

of observed IOD events for both the developing and decaying years. The model was integrated for 42 years, and the first 4 years were discarded given that it takes a few years to reach the model equilibrium state. Since the SSTA forcing has a 2-year period, i.e., the developing and decaying years of the IOD, we have 19 ensemble members available for analysis.

We also used a linear ocean model[61,62] to examine the impact of wind anomalies induced by the IOD on the tropical Atlantic Ocean. The ocean model was forced by differences of surface wind stress anomalies during June–November between the two sets of AGCM experiments. It has 25 baroclinic modes, and a horizontal resolution of 0.5°. The model experiment started from a state of rest with a realistic background stratification and a flat ocean bottom with a depth of 4000 m. It was spun up for 5 years, and integrated forward for another 5 years, which are analyzed in this study.

Given that local air-sea interaction plays a crucial role in the formation of the Atlantic Niño, we further performed pacemaker experiment using the NCAR

CESM1, in which we restored the Indian Ocean SSTA to observed values (Supplementary Fig. 10). A sponge layer was applied at the western, southern, and eastern boundaries. The simulation period is 1920–2019, and we analyzed the results during 1950–2019. The model was forced with the same external forcing (both anthropogenic and natural) as those applied in the historical simulation (1920–2005) and Representative Concentration Pathway 8.5 (RCP8.5) future projection (2006–2019) from Coupled Model Intercomparison Project Phase 5 (CMIP5). We obtained a ten-member ensemble by slightly perturbing the initial conditions, and the ensemble averaged results isolate the effect of the Indian Ocean SSTA on global climate variability and change. Hence, the Atlantic SSTA in the pacemaker experiment is solely caused by the Indian Ocean SSTA effect, and, therefore, we consider an Atlantic Niño event associated with the positive IOD forcing as long as the DMI exceeds one standard deviation within 5 months prior to the Atlantic Niño peak in the model. Similarly for the Atlantic Niña and negative IOD (e.g., Supplementary Fig. 12).

## Data availability

The HadISST data set is available at https://www.metoffice.gov.uk/hadobs/hadisst/data/download.html. The ERA-20C data set can be downloaded from https://www.ecmwf.int/en/forecasts/datasets/reanalysis-datasets/era-20c. The ORAS4 data is available at https://www.cen.uni-hamburg.de/icdc/data/ocean/easy-init-ocean/ecmwf-ocean-reanalysis-system-4-oras4.html. The satellite-derived daily sea level data can be downloaded from https://cds.climate.copernicus.eu/cdsapp#!/dataset/satellite-sea-level-global?tab=overview. The CESM large ensemble results are available at https://www.cesm.ucar.edu/projects/community-projects/LENS/data-sets.html. The scripts used to analyze data and the numerical model results in this study are available from the corresponding author upon request.

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

## Acknowledgements

L.Z. and W.H. are supported by NOAA CPO NA20OAR4310480, NSF AGS 1935279, and NASA OSTST 80NSSC21K1190. We would like to acknowledge high-performance computing support from Cheyenne (doi:10.5065/D6RX99HX) provided by NCAR's Computational and Information Systems Laboratory, sponsored by the National Science Foundation.

## Author contributions

L.Z. and W.H. designed the study and wrote the paper. L.Z. conducted analysis, carried out numerical model experiments, and produced figures.

## Competing interests

The authors declare no competing interests.
