## [Peer Review File · Nature Communications]

REVIEWER COMMENTS

Reviewer #1 (Remarks to the Author):

Summary

The authors examine the potential influence of positive Indian Ocean Dipole (IOD) events on the development of Atlantic Niños in the following months. Based on observations and GCM experiments they suggest that IOD events tend to lead to equatorial Atlantic SST anomalies of the same sign about 5 months later.

The manuscript is well written and the results are novel and of wider interest to the community. It will be important, however, to quantify the importance of the IOD influence on the equatorial Atlantic to get an estimate of the relative importance of this linkage. There should also be some more analysis and discussion of possible caveats, such as the fact that equatorial Atlantic SST anomalies already exist during the onset of the IOD, and the potential influence of ENSO on both the IOD and Atlantic Niños.

Major Comments

Figure 1 shows that there are already SST anomalies in the equatorial Atlantic during the developing phase of the IOD. How do the authors interpret this? Could there be a third process that influences both?

While it has been argued that the Atlantic Niño as well as the IOD may be partially independent of ENSO, it is conceivable that both events could be influenced by preceding ENSO events. Could this be a factor here? Case in point is the 1998 Atlantic Niño, which followed a pIOD event but also one of the strongest El Niño events on record. While the authors attempt to remove the ENSO influence through partial regression, this may not be 100% successful.

It seems that there is no seasonal stratification performed on the observational analysis (Figs. 1-3). Both the Atlantic Niño and the IOD, however, have strong seasonal preferences. How does the partial regression change when the authors key it on boreal fall, the preferred season of the IOD? So, for each year, you calculate the SON mean of the IOD and regress October (Nov, Dec, etc.) SST of each year on this.

The suggested sequence of events is intriguing but I am wondering how well this fits with the evolution of individual events. Taking 2019 as an example, the Atlantic Niño developed from October, which roughly coincides with the peak of the IOD. This is different from the ~5 month lag suggested by Fig. 1. It would be interesting to check the evolution of IOD and Atlantic Niño for the 4 years identified by the authors (l. 155 ff.). Figure S7 shows a composite of the 4 years but it would also be instructive to look at individual years. Also, please state the 4 pIOD and 4 nIOD events in question in the main text.

It is important to put into context the strength of the suggested IOD influence on the equatorial Atlantic. For the partial regression analysis, what is the strength of typical IOD events, and how do the regression values in the tropical Atlantic (multiplied by the strength of IOD events) compare to those of typical Atlantic Niños? Based on this, are IODs a major influence or just a minor contribution? Are the regressions sensitive to the extreme IOD of 2019?

For the AGCM experiments, how do the wind anomalies over the tropical Atlantic compare to those for typical Atlantic Niños? (since you are using pIOD minus nIOD, all values should be divided by half)

Finally, how do the SSH anomalies in the simple ocean model compare to those during typical Atlantic Niño events in observations or reanalysis?

Minor Comments

I. 28: I believe the reference Lübbecke (2013) may be better placed on line 55, as it is about the competing effects of Rossby/Kelvin waves vs. meridional advection.

I. 30: There is a more recent review paper by Richter and Tokinaga (2020) that could be cited here.

II. 34-35: You could also cite Pottapinjara et al. (2016) here.

II. 37-38: Okumura and Xie (2006; cited on the following line) suggest that the late fall Atlantic Niño is separate from the summer one and name it Atlantic Niño II.

II. 56-57: Westerly wind anomalies in MAM typically are a large factor in the development of warm events, as shown, e.g., in Lübbecke et al. (2018) and Richter and Tokinaga (2020). This is analogous to westerly wind bursts in the tropical Pacific.

II. 108-112: The influence of along-shore wind anomalies in the generation of coastal warm events (Benguela Niños) is somewhat controversial. Polo et al. (2008) and Richter et al. (2010) argue for an important contribution, while Florenchie et al. (2003) and Bachelery et al. (2016) argue for a minor or no contribution from local winds and attribute events remote forcing from the equatorial Atlantic.

Please clarify the setup of the AGCM experiments. Are SSTs set to annual climatology everywhere except the tropical Indian Ocean?

II. 139-140

"Interbasin interactions play a profound role in global climate variability and change, and has drawn considerable attentions recently."

Grammar and style issues. Suggest: "Interbasin interactions play a profound role in global climate variability and change, and have attracted considerable attention recently."

II. 203-204: "ECHAM4.6 from Max Planck Institute (MPI) in Hamburg (HAM), which is a branch from the ECMWF"

I believe this is not correct. The ECMWF model (IFS) was developed at ECMWF in collaboration with Meteo France.

References

Bachelery, M.-L., Illig, S., and Dadou, I. (2016), Interannual variability in the South-East Atlantic Ocean, focusing on the Benguela Upwelling System: Remote versus local forcing, *J. Geophys. Res. Oceans*, 121, 284– 310, doi:10.1002/2015JC011168.

Florenchie, P., J. R. E. Lutjeharms, C. J. C. Reason, S. Masson, and M. Rouault (2003), The source of Benguela Niños in the South Atlantic Ocean, *Geophys. Res. Lett.*, 30(10), 1505, doi:10.1029/2003GL017172.

Polo, I., A. Lazar, B. Rodriguez-Fonseca, and S. Arnault (2008), Oceanic Kelvin waves and tropical Atlantic intraseasonal variability: 1. Kelvin wave characterization, *J. Geophys. Res.*, 113, C07009, doi:10.1029/2007JC004495.

Pottapinjara, V., M. S. Girishkumar, S. Sivareddy, M. Ravichandran, and R. Murtugudde, 2016: Relation between the upper ocean heat content in the equatorial Atlantic during boreal spring and the Indian monsoon rainfall during June–September. *Int. J. Climatol.*, 36, 2469–2480, <https://doi.org/10.1002/joc.4506>.

Richter, I., Behera, S. K., Masumoto, Y., Taguchi, B., Komori, N., and Yamagata, T. (2010), On the triggering of Benguela Niños: Remote equatorial versus local influences, *Geophys. Res. Lett.*, 37, L20604, doi:10.1029/2010GL044461.

Richter, I., and H. Tokinaga, 2020: The Atlantic Niño: Dynamics, thermodynamics, and teleconnections. *Tropical and Extratropical Air–Sea Interactions*, S. K. Behera, Ed., Elsevier, 171–205, <https://doi.org/10.1016/B978-0-12-818156-0.00008-3>.

Reviewer: Ingo Richter

Reviewer #2 (Remarks to the Author):

The study uses observations, reanalysis and model experiments (AGCM and CESM-LE) to show that positive Indian Ocean Dipole (IOD) can trigger Atlantic Niño events. The IOD affects the tropical Atlantic via a Gill-type response in the atmosphere where enhanced rainfall and diabatic heating in the western IOD pole lead to Rossby waves to the tropical South Atlantic, which weakens the Atlantic trade winds generating a subsequent oceanic response similar to El Niño. I find the manuscript clear, interesting, and well written. Tropical basin interaction is a topic of great interest to the climate community, and the links between the Indian Ocean and the Atlantic have not been explored much in the literature. This study thus covers some of the gaps in the Indian Ocean-Atlantic interactions and is likely to spark the interest of the community and provide ideas for follow up studies.

My main comment relates to the timing of the Atlantic Niño. Atlantic Niño peaks in JJA. The analysis shown in the manuscript suggests that the peak of the Atlantic Niño occurs ~5 months after the peak of the IOD (Fig.1). Given IOD matures in SON, this means that IOD-induced Atlantic Niño tends to develop and peak earlier (~April) than its observed peak. If that is correct, it is worth adding a discussion in the text. The equatorial Atlantic is a narrow basin, and equatorial waves should propagate relatively fast -- downwelling Kelvin waves cited in L.102 (waves are not clear to me in Fig. S4, consider adding thin dashed lines for reference) – making an earlier appearance of Atlantic Niños.

On the predictability note (L.155-156), only 28% (25%) of the observed Atlantic Niño (Niña) events are preceded by positive (negative) IOD. Given the low sample size in the historical record, and I am not convinced that the findings of this study can effectively improve the predictability of the Atlantic Niño. Can IOD solely trigger Atlantic Niños, or it contributes to amplify processes initiated by the Pacific variability and/or local processes? Perhaps when combined with the Pacific variability, the IOD can provide enhanced predictability?

L.167: El Niño.

Reviewer #3 (Remarks to the Author):

Review of manuscript NCOMMS-21-06746 “Indian Ocean Dipole leads to Atlantic Niño” by Zhang and Han, submitted to *Nature Communications*.

This paper explores the interaction between the Indian and Atlantic tropical ocean basins, focusing in the impact of the Indian Ocean Dipole (SON) in the development of the Atlantic Niño in the following season (JJA). The results show that the IOD induces westerly wind anomalies in the equatorial Atlantic that trigger the Atlantic Niño.

Although the result is novel and promising, I don't think it can be presented in a *Nature Communications* paper in its present form. The main caveat of the work is the use of an AGCM plus an ocean linear model instead of a pacemaker experiment with full coupled model in the Atlantic and Pacific, restored to an IOD pattern over the Indian Ocean. The present experiments can't show the evolution of the Atlantic from SON to the next JJA, and so it is not possible to

assure that the atmospheric anomalies shown in SON in the equatorial Atlantic would be able to reinforce themselves throughout the whole winter and spring and lead to an Atlantic Niño in summer.

I would therefore recommend to perform a pacemaker experiment and then resubmit the paper.

Below I show some remarks that the authors should also take into account before resubmission.

I have doubts about the lead-lag timing of the figures. If the IOD is calculated in SON, then the lag "1 mon" would be OND, lag "3 mon" would be "DJF", lag "5 mon" would be "FMA" and lag "7 mon" would be "AMJ". According to this, the Atlantic Niño peak appears in spring, but we know that in reality the Atlantic Niño shows a peak in summer and a second one in late fall-early winter. Please explain this inconsistency between your results and the literature.

Although the impact of the IOD in the equatorial Atlantic is plausible, the presence of warm anomalies in the equatorial Atlantic from lag -5 in figure 1 cannot be overlooked, as can be the trigger for the Atlantic anomalies by themselves.

Lines 82-86: I don't see a clear response of the Southern Atlantic Anticyclone in figure 1. Maybe if you show a broader latitudinal band (at least -40 to 40) or the streamfunction anomalies, the Gill type response will be clearer.

Figure S4: I can't see clear propagation of the Kelvin waves in any of the panels. Moreover, 1995 and 1998 are positive Atlantic Niños and have more or less opposite patterns of sea level anomalies in figure S4. Could you please mark the waves in the figure and explain better the differences?

Response to Reviews

We thank the three Reviewers for carefully reading the manuscript and providing constructive comments, which has led to improvement of the manuscript. Below, we respond to each reviewer's comments sequentially, repeating each comment in "black". Our responses are in blue, and we have put all supporting figures at the end of this response file.

Reviewer #1

The authors examine the potential influence of positive Indian Ocean Dipole (IOD) events on the development of Atlantic Niños in the following months. Based on observations and GCM experiments they suggest that IOD events tend to lead to equatorial Atlantic SST anomalies of the same sign about 5 months later.

The manuscript is well written and the results are novel and of wider interest to the community. It will be important, however, to quantify the importance of the IOD influence on the equatorial Atlantic to get an estimate of the relative importance of this linkage. There should also be some more analysis and discussion of possible caveats, such as the fact that equatorial Atlantic SST anomalies already exist during the onset of the IOD, and the potential influence of ENSO on both the IOD and Atlantic Niños.

We appreciate the reviewer's nice summary of this work and the positive comments, which are encouraging. Below we have carefully addressed each of the reviewer's concerns/comments, and revised the manuscript accordingly.

Major Comments:

Figure 1 shows that there are already SST anomalies in the equatorial Atlantic during the developing phase of the IOD. How do the authors interpret this? Could there be a third process that influences both?

While it has been argued that the Atlantic Niño as well as the IOD may be partially independent of ENSO, it is conceivable that both events could be influenced by preceding ENSO events. Could this be a factor here? Case in point is the 1998 Atlantic Niño, which followed a pIOD event but also one of the strongest El Niño events on record. While the authors attempt to remove the ENSO influence through partial regression, this may not be 100% successful.

We appreciate the constructive comment. We also note the pre-existing warm SSTAs in the tropical Atlantic Ocean (e.g., Fig. 1b of the original manuscript), when the IOD is still at its initial developing stage. The existence of these Atlantic signals is mainly due to the partial regression approach that we used to remove the ENSO influence. Because El Niño tends to induce cold SSTAs in the tropical Atlantic (Fig. S1), removing those associated signals introduce artificial warm SSTAs hence the pre-existing warm Atlantic SSTAs. Indeed, the lead-lag regression of total SSTAs (without removing ENSO-related signals) on the DMI (Fig. R1) shows that the tropical Atlantic Ocean is dominated by negative SSTAs from -7 to -3 month, with very

weak warm SSTAs that are statistically insignificant. Given that our results are similar with and without ENSO influences, we have decided to not remove ENSO-associated signals in the revised manuscript to avoid confusion (e.g., replace original Fig. 1 with Fig. R1).

We also agree that while the regression analysis suggests that ENSO may not yield significant influences on Atlantic Niño in general, it could still play a significant role in certain years, which may affect the suggested IOD effect on Atlantic Niño in this study. To test this possibility, we have examined each of the selected pIOD-Atlantic Niño as well as nIOD-Atlantic Niña years. Please see below for the results and more detailed discussions.

It seems that there is no seasonal stratification performed on the observational analysis (Figs. 1-3). Both the Atlantic Niño and the IOD, however, have strong seasonal preferences. How does the partial regression change when the authors key it on boreal fall, the preferred season of the IOD? So, for each year, you calculate the SON mean of the IOD and regress October (Nov, Dec, etc.) SST of each year on this.

We agree that it is important to consider seasonality in our analysis, since the evolutions of both IOD and Atlantic Niño exhibit strong seasonal dependence. Following the reviewer's suggestion, we have calculated the lead-lag regression of 3-month mean Atlantic and Indian Ocean SSTAs on the SON-averaged DMI (Fig. R2). We note that statistically significant tropical Atlantic warm SSTAs start to appear after the IOD peak, then further develop, and reach the peak 2-4 months later. These results are consistent with our regression results using monthly data (cf. Fig. 1), which shows 3 to 5-month lead of pIOD peak over the Atlantic Niño peak. This is an important point, which we have explicitly discussed in the revised manuscript (L92-104).

The suggested sequence of events is intriguing but I am wondering how well this fits with the evolution of individual events. Taking 2019 as an example, the Atlantic Niño developed from October, which roughly coincides with the peak of the IOD. This is different from the ~5 month lag suggested by Fig. 1. It would be interesting to check the evolution of IOD and Atlantic Niño for the 4 years identified by the authors (l. 155 ff.). Figure S7 shows a composite of the 4 years but it would also be instructive to look at individual years. Also, please state the 4 pIOD and 4 nIOD events in question in the main text.

We appreciate the constructive comment. In the original manuscript, we define the Atlantic Niño as the years when the JJA-mean ATL3 index exceeds one standard deviation, and similarly for pIOD years when the SON-mean DMI is higher than one standard deviation. However, we also note that although the Atlantic Niño generally tends to occur in boreal summer, many of them occur in other seasons. As a result, the previous definition may miss some Atlantic Niño events, including the 2019 super Atlantic Niño event that peaks in December 2019-January 2020. Hence, in the revised manuscript, we have modified our definition of Atlantic Niño events, which are now defined based on whether the peak of 3-month smoothed ATL3 index exceeds one standard deviation. For pIOD events, we used a similar definition using 3-month smoothed DMI. Based on the new criteria, we have selected 33 Atlantic Niño and 34 Atlantic Niña events during 1950-2019. Furthermore, if an Atlantic Niño is preceded by a pIOD within 5 months prior to its peak, the Atlantic Niño is considered associated with pIOD forcing. We have selected 10 such Atlantic Niño events (30%), and similarly 10 Atlantic Niña events associated with nIOD (29%). The

composited differences between the Atlantic Niño and Niña events that are associated with IOD are shown in Fig. R3, which show that the peak of pIOD precedes the Atlantic Niño by 2-4 months. In the regression analysis results (Fig. 1), the equatorial Atlantic warm SSTAs are strong in 3-5 months after the IOD peak, and therefore the two analyses are generally consistent.

We further analyzed evolutions of DMI for each selected Atlantic Niño-pIOD and Niña-nIOD events (years also listed in the figure), as suggested by the reviewer in Fig. R4, which shows DMI from 5 months prior to the Atlantic Niño/Niña peak (month 0). Note that we have subtracted the values of DMI in month -5 from all the DMIs, since we mainly focus on their evolutions. As such, a persistent decrease in the DMI in Fig. R4a suggests that the pIOD reaches its peak 5 months prior to the Atlantic Niño. Of total 10 events, there are 4 pIODs that peak in month -5, 2 peak in month -4, 2 peak in month -3, 2 peak in month -2 (including the 2019 event). DMIs during the 10 Atlantic Niña-nIOD events are shown in Fig. R4b, and a persistent increase now means the nIOD peaks in month -5, and there are 3 such events, with the rest of them peaking in month -1 to -4. Hence, the event-to-event differences in the lead time of IOD over Atlantic Niño are quite significant. This is because the IOD acts as a trigger for the Atlantic Niño, which then further develops through large-scale air-sea interactions, while other factors (such as extratropical forcing) may sometimes also play a role. The spread across different events also explains the slightly different values in the lead time through regression on monthly DMI (3-5 months), composite analysis (2-4 months) and regression on SON-mean DMI (2-4 months). This is an important point, and we appreciate the reviewer for bringing it up. We have revised manuscript for clarifications (L92-104, 220-235).

In addition, as discussed in more details below, we also performed the so-called pacemaker experiments and analyzed the 2019 IOD and Atlantic Niño (Fig. R5). The model results indeed show that the IOD causes the Atlantic Niño in 2019 with a 2-month lead. This further supports our proposed mechanism.

It is important to put into context the strength of the suggested IOD influence on the equatorial Atlantic. For the partial regression analysis, what is the strength of typical IOD events, and how do the regression values in the tropical Atlantic (multiplied by the strength of IOD events) compare to those of typical Atlantic Niños? Based on this, are IODs a major influence or just a minor contribution? Are the regressions sensitive to the extreme IOD of 2019?

For the AGCM experiments, how do the wind anomalies over the tropical Atlantic compare to those for typical Atlantic Niños? (since you are using pIOD minus nIOD, all values should be divided by half)

Finally, how do the SSH anomalies in the simple ocean model compare to those during typical Atlantic Niño events in observations or reanalysis?

We agree that evaluating quantitative contribution of the pIOD forcing to the Atlantic Niño is important, and we thank the reviewer for this careful thought. To assess the strength of typical IOD events, we calculated the standard deviation of SON-mean DMI, which is 0.38 °C. The standard deviation of JJA-mean ATL3 is 0.51 °C. The maximum regression coefficient in Fig. R2 averaged over the ATL3 region (regression on SON-mean DMI) is 0.37 °C/°C. This simple calculation suggests that the IOD forcing may contribute to 27% ($0.38 \times 0.37 / 0.51$) of the Atlantic Niño SSTAs, which is slightly smaller but close to the percentage of Atlantic Niño/Niña events that are associated with the IOD forcing (~30%). Besides, exclusion of the 2019 extreme IOD

event, which is followed by a super Atlantic Niño event, does not affect our results. In fact, Fig. R2 shows the lead-lag regression on SON-mean DMI, which has already excluded the year 2019 because the SST data we use ends in Dec 2019.

Furthermore, although results from both AGCM and ocean model may help prove our hypothesis that the IOD may affect the Atlantic Niño through the atmospheric bridge, we also would like to note that the two models do not include air-sea interaction processes, whereas the Bjerknes feedback plays an important role in amplifying both wind and SST/SSH anomalies. Hence, we do not expect that the standalone atmosphere or ocean model results duplicate observations, but the modeled winds, currents and SSHA patterns do bear resemblance with observed ones (compare the equatorial westerlies of Figs. 1e-1f with Figs. 4a-4b, and current+SSH anomaly patterns of Figs. 3e-3f with Fig. S9).

Given that air-sea interaction plays a crucial role in the growth of Atlantic Niño/Niña to its mature stage after the IOD triggers the Atlantic SSTA, direct comparisons of IOD & Atlantic Niño/Niña amplitudes may not be a good measure of IOD's contribution. Indeed, in response to Reviewer #3's comments, we have performed the pacemaker experiments, in which we have restored SSTAs in the Indian Ocean to the observed values while leave the atmosphere and ocean fully coupled elsewhere (Fig. R6). Hence, positive air-sea feedback has been fully considered outside the Indian Ocean in this model setup, which allows us to compare the number and amplitude of Atlantic Niño events that are and are not associated with IOD forcing (Fig. R7). The ensemble mean results from pacemaker experiment show that the IOD-induced Atlantic Niño/Niña events are slightly stronger than the IOD-independent events. Meanwhile, there are 31%–33% of total Atlantic Niño/Niña that are forced by the IOD, which also agrees with observational results.

These results together suggest that the IOD forcing does play an important role in forcing the Atlantic Niño events, and it contributes to around one third of them. Given that the Atlantic Niño may be affected by many other factors (including the AMM), this contribution is not small. We have added the discussions and new results above in the revised manuscript (L149-162, 181-189). We appreciate the reviewer's constructive comment, which motivates us to perform additional analysis that has helped improve this study.

Minor Comments:

I. 28: I believe the reference Lübbecke (2013) may be better placed on line 55, as it is about the competing effects of Rossby/Kelvin waves vs. meridional advection.

I. 30: There is a more recent review paper by Richter and Tokinaga (2020) that could be cited here.

II. 34-35: You could also cite Pottapinjara et al. (2016) here.

II. 37-38: Okumura and Xie (2006; cited on the following line) suggest that the late fall Atlantic Niño is separate from the summer one and name it Atlantic Niño II.

II. 56-57: Westerly wind anomalies in MAM typically are a large factor in the development of warm events, as shown, e.g., in Lübbecke et al. (2018) and Richter and Tokinaga (2020). This is analogous to westerly wind bursts in the tropical Pacific.

II. 108-112: The influence of along-shore wind anomalies in the generation of coastal warm

events (Benguela Niños) is somewhat controversial. Polo et al. (2008) and Richter et al. (2010) argue for an important contribution, while Florenchie et al. (2003) and Bachelery et al. (2016) argue for a minor or no contribution from local winds and attribute events remote forcing from the equatorial Atlantic.

Please clarify the setup of the AGCM experiments. Are SSTs set to annual climatology everywhere except the tropical Indian Ocean?

ll. 139-140

“Interbasin interactions play a profound role in global climate variability and change, and has drawn considerable attentions recently.”

Grammar and style issues. Suggest: “Interbasin interactions play a profound role in global climate variability and change, and have attracted considerable attention recently.”

ll. 203-204: “ECHAM4.6 from Max Planck Institute (MPI) in Hamburg (HAM), which is a branch from the ECMWF”

I believe this is not correct. The ECMWF model (IFS) was developed at ECMWF in collaboration with Meteo France.

Thanks for the careful reading and the helpful comments. The text has been revised as suggested.

Reviewer #2

The study uses observations, reanalysis and model experiments (AGCM and CESM-LE) to show that positive Indian Ocean Dipole (IOD) can trigger Atlantic Niño events. The IOD affects the tropical Atlantic via a Gill-type response in the atmosphere where enhanced rainfall and diabatic heating in the western IOD pole lead to Rossby waves to the tropical South Atlantic, which weakens the Atlantic trade winds generating a subsequent oceanic response similar to El Niño.

I find the manuscript clear, interesting, and well written. Tropical basin interaction is a topic of great interest to the climate community, and the links between the Indian Ocean and the Atlantic have not been explored much in the literature. This study thus covers some of the gaps in the Indian Ocean-Atlantic interactions and is likely to spark the interest of the community and provide ideas for follow up studies.

We appreciate the reviewer's encouraging comment. The two main comments below have helped improve this study, based on which we have carefully revised the manuscript.

My main comment relates to the timing of the Atlantic Niño. Atlantic Niño peaks in JJA. The analysis shown in the manuscript suggests that the peak of the Atlantic Niño occurs ~5 months after the peak of the IOD (Fig.1). Given IOD matures in SON, this means that IOD-induced Atlantic Niño tends to develop and peak earlier (~April) than its observed peak. If that is correct, it is worth adding a discussion in the text. The equatorial Atlantic is a narrow basin, and equatorial waves should propagate relatively fast -- downwelling Kelvin waves cited in L.102 (waves are not clear to me in Fig. S4, consider adding thin dashed lines for reference) – making an earlier appearance of Atlantic Niños.

We appreciate the constructive comment. First, we would like to note that our original definitions of Atlantic Niño and IOD based on whether the seasonal mean ATL3 (in summer) and DMI (in fall) exceed their respective standard deviations have been modified. We now use 3-month smoothed indices to define the two climate events as when their peaks exceed one standard deviation. The reason that we change the definitions is because although most of the Atlantic Niño events tend to occur in boreal summer, many of them occur in other seasons. As a result, the previous definition may miss many Atlantic Niño events, including the 2019 super Atlantic Niño event that peaks in December 2019-January 2020. With the new definitions, we further define Atlantic Niño events that are associated with the pIOD forcing as those that are preceded by pIOD events within 5 months prior to their peaks. Similar for the nIOD-dependent Atlantic Niña. Using the new definitions, we have selected 33 Atlantic Niño during 1950-2019, 10 of which are associated with pIOD (30%). Similarly, 10 of 34 Atlantic Niña events are associated with nIOD (29%). The composited differences between the Atlantic Niño and Niña events that are associated with IOD are shown in Fig. R3, which show that the peak of pIOD precedes the Atlantic Niño by 2-4 months. In the regression analysis results (Fig. 1), the equatorial Atlantic warm SSTAs are strong in 3-5 months after the IOD peak, and therefore the two analyses are generally consistent.

Then, we examine the timing of the Atlantic Niño by comparing the peak month of the Atlantic Niño events between those are and are not associated with the pIOD forcing (Fig. R8). Results indeed exhibit prominent differences between the two categories: While the independent Atlantic Niño tends to peak in boreal summer, more than half of the pIOD-associated Atlantic Niño

events occur between November and January. Similar for Atlantic Niña and nIOD. Hence, the IOD does seem to affect the timing of Atlantic Niño.

We further analyzed the evolution of the DMI prior to the Atlantic Niño peak for each selected event (Fig. R4). Results show that although the regression and composite analyses show 2 to 5-month lead of the DMI over ATL3, the time lead may vary from case to case. Note that we have subtracted the values of DMI in month -5 from all the DMIs in Fig. R4, since we mainly focus on their evolutions. As a result, a persistent decrease in the DMI in Fig. R4a suggests that the pIOD reaches its peak 5 months prior to the Atlantic Niño. Of total 10 events, there are 4 pIODs that peak in month -5 , 2 peak in month -4 , 2 peak in month -3 , 2 peak in month -2 (including the 2019 event). DMIs during the 10 Atlantic Niña-nIOD events are shown in Fig. R4b, and a persistent increase now means the nIOD peaks in month -5 , and there are 3 such events, with the rest of them peaking in month -1 to -4 . Hence, the event-to-event differences in the lead time of IOD over Atlantic Niño are quite significant. This is because the IOD acts as a trigger for the Atlantic Niño, which then further develops to its peak stage through large-scale air-sea interactions, while other factors (such as extratropical forcing) may sometimes also play a role. This is an important point, and we appreciate the reviewer for bringing it up. We have revised the manuscript carefully to address this issue (L92-104, 181-194, 220-235). We have also revised Fig. S4 of the original manuscript by adding arrows that represent Kelvin wave propagation (Fig. S7 of the revised manuscript).

On the predictability note (L.155-156), only 28% (25%) of the observed Atlantic Niño (Niña) events are preceded by positive (negative) IOD. Given the low sample size in the historical record, and I am not convinced that the findings of this study can effectively improve the predictability of the Atlantic Niño. Can IOD solely trigger Atlantic Niños, or it contributes to amplify processes initiated by the Pacific variability and/or local processes? Perhaps when combined with the Pacific variability, the IOD can provide enhanced predictability?

We appreciate the thoughtful comment. Our results suggest that the IOD forcing contributes to around one third of the Atlantic Niño events. Given that the Atlantic Niño may be affected by many other factors (including the AMM), this contribution is not small, especially for the events that occur in boreal fall and winter seasons. From the predictability point of view, our results help explain an important source of forcing for predicting Atlantic Niño/Niña events that occur in boreal fall-winter seasons.

In addition, in response to Reviewer #3's comments, we have performed the so-called pacemaker experiments, in which we restore SSTAs in the Indian Ocean to the observed values while leave the atmosphere and ocean fully coupled elsewhere. We have ten ensemble members by slightly perturbing the initial conditions. Hence, the ensemble averaged results isolate the effect of the Indian Ocean on climate variability in other regions, while other influences including the Pacific ENSO forcing have been filtered out. This new set of model experiments allows us to more cleanly examine the effect of the IOD on Atlantic Niño, and it shows that there are 31%–33% of total Atlantic Niño/Niña that are forced by the IOD, which also agrees with observational results.

Since both the new observational analysis and pacemaker experiments suggest a prominent impact of IOD on Atlantic Niño, we think that the finding presented in this study provides a new source of predictability and therefore it may potentially help improve predictability of the Atlantic Niño. We have added discussions above in the revised manuscript and made

clarifications as well (L149-162,181-189).

L.167: El Niño.

The text has been revised.

Reviewer #3

Review of “Indian Ocean Dipole leads to Atlantic Niño” by Zhang and Han.

This paper explores the interaction between the Indian and Atlantic tropical ocean basins, focusing on the impact of the Indian Ocean Dipole (SON) in the development of the Atlantic Niño in the following season (JJA). The results show that the IOD induces westerly wind anomalies in the equatorial Atlantic that trigger the Atlantic Niño.

Although the result is novel and promising, I don't think it can be presented in a Nature Communications paper in its present form. The main caveat of the work is the use of an AGCM plus an ocean linear model instead of a pacemaker experiment with full coupled model in the Atlantic and Pacific, restored to an IOD pattern over the Indian Ocean. The present experiments can't show the evolution of the Atlantic from SON to the next JJA, and so it is not possible to assure that the atmospheric anomalies shown in SON in the equatorial Atlantic would be able to reinforce themselves throughout the whole winter and spring and lead to an Atlantic Niño in summer.

I would therefore recommend to perform a pacemaker experiment and then resubmit the paper.

We appreciate the reviewer's constructive comment, and agree that performing additional pacemaker experiments that allow atmosphere-ocean coupling would provide useful insights, since positive air-sea feedback plays a crucial role in the development of Atlantic Niño. In response to this comment, we have performed such experiments using NCAR CESM1, in which we have restored the Indian Ocean SSTAs to observed values (Fig. R6). The simulation period is 1920-2019, and we have ten ensemble members with slightly perturbed initial conditions. The ensemble mean results therefore isolate the effect of Indian Ocean on global climate variability.

Regression results from the pacemaker experiments show similar patterns to those in observations (Fig. R9). Prior to the IOD peak, both the positive rainfall anomalies over the tropical Indian Ocean and the tropical Atlantic Ocean wind anomalies are weak (Fig. R9a). As the IOD develops, rainfall increases off the east coast of Africa, and correspondingly westerly wind anomalies start to appear and strengthen over the tropical Atlantic Ocean, which warm the underlying ocean surface (Fig. R9b-R9d). During the IOD decaying phase, Indian Ocean rainfall anomalies significantly weaken, while the SST and wind anomalies associated with the Atlantic Niño sustain and further strengthen even 3 months after the IOD peak (Fig. R9e). In month 5, IOD anomalies almost disappear, and Atlantic Niño also weakens in the model (Fig. R9f).

The pacemaker experiments clearly show the atmospheric teleconnection through which the IOD affect the tropical Atlantic winds and the formation of Atlantic Niño, which are very helpful, and we have added results and discussions above in the revised manuscript (L149-162, 266-274). Please also see below for detailed discussion on the effect of IOD on the timing of Atlantic Niño.

Below I show some remarks that the authors should also take into account before resubmission.

I have doubts about the lead-lag timing of the figures. If the IOD is calculated in SON, then the lag “1 mon” would be OND, lag “3 mon” would be “DJF”, lag “5 mon” would be “FMA” and lag “7 mon” would be “AMJ”. According to this, the Atlantic Niño peak appears in spring, but we know that in reality the Atlantic Niño shows a peak in summer and a second one in late fall-early winter. Please explain this inconsistency between your results and the literature.

This is an important point. In the regression analysis, we used monthly DMI throughout the entire analysis period, instead of using SON-mean DMI. As a result, seasonality is not considered in Fig. 1. In response to the reviewer's comment, we have calculated lead-lag regression on SON-mean DMI, in which month 1 means OND anomalies (Fig. R2). The results show negligible SSTAs in the tropical Atlantic Ocean prior to the IOD peak, and a peak of Atlantic Niño in 2-4 months after the IOD peak, which is similar to the regression using monthly data (3-5 months).

Note that in the revised MS, our original definitions of the Atlantic Niño and the IOD based on whether the seasonal mean ATL3 (in summer) and DMI (in fall) exceed their respective standard deviations have been modified. We now use 3-month smoothed indices to define the two climate events as when their peaks exceed one standard deviation. The reason that we change their definitions is because although the Atlantic Niño generally tends to peak in boreal summer, many of them peak in other seasons. As a result, the previous definitions may miss many Atlantic Niño events, including the 2019 super Atlantic Niño event that peaks in December 2019-January 2020, which followed an unprecedented positive IOD event that peaks in October 2019. Composite results using the new definitions agree with the regression results well (Fig. R3), which suggest that the IOD leads Atlantic Niño by 2-4 months. In the regression analysis results (Fig. 1), the equatorial Atlantic warm SSTAs are strong in 3-5 months after the IOD peak, and therefore the two analyses are generally consistent.

The time lead of IOD over the Atlantic Niño, however, may vary significantly from case to case (Fig. R4). The evolutions of the DMIs prior to the Atlantic Niño peaks show that of total 10 events identified in observations, there are 4 pIODs that peak in month -5, 2 peak in month -4, 2 peak in month -3, 2 peak in month -2 (including the 2019 event). During the 10 Atlantic Niña-nIOD events, 3 nIOD events peak in month -5, and the rest of them peak in month -1 to -4. Hence, the event-to-event differences in the lead time of IOD over Atlantic Niño are quite significant. This is because the IOD acts as a trigger for the Atlantic Niño, which then further develops through large-scale air-sea interactions, while other factors (such as extratropical forcing) may sometimes also play a role. The spread across different events also explains the slightly different values in the lead time through regression on monthly DMI (3-5 months), composite analysis (2-4 months) and regression on SON-mean DMI (2-4 months). This is an important point, and we appreciate the reviewer for bringing it up. We have revised manuscript for clarifications (L92-104, 220-235).

We further found that while the independent Atlantic Niño tends to peak in boreal summer, more than half of the pIOD-associated Atlantic Niño events occur between November and January, just a few months after the IOD peak (Fig. R8). Similar for the effect of negative IOD on Atlantic Niña. Hence, the IOD does seem to affect the timing of Atlantic Niño. It is indeed very important to analyze in detail how the IOD affects the timing of Atlantic Niño, and we have revised the manuscript based on the new results and discussions above (L190-194).

Although the impact of the IOD in the equatorial Atlantic is plausible, the presence of warm anomalies in the equatorial Atlantic from lag -5 in figure 1 cannot be overlooked, as can be the trigger for the Atlantic anomalies by themselves.

We appreciate the comment, and we also note the pre-existing warm SSTAs in the tropical Atlantic Ocean (e.g., Fig. 1b), when the IOD is still at its initial developing stage. The existence

of these Atlantic signals is mainly due to the partial regression approach that we used to remove the ENSO influence. Because El Niño tends to induce cold SSTAs in the tropical Atlantic (Fig. S1), removing those associated signals introduce artificial warm SSTAs hence the pre-existing warming Atlantic SSTAs. Indeed, the lead-lag regression of total SSTAs (without removing ENSO-related signals) on the DMI (Fig. R1) shows that the tropical Atlantic Ocean is dominated by negative SSTAs from -7 to -3 month, with very weak warm SSTAs that are statistically insignificant. Given that our results are similar with and without ENSO influences, we have decided to not remove ENSO-associated signals in the revised manuscript to avoid confusion (e.g., replace original Fig. 1 with Fig. R1).

Lines 82-86: I don't see a clear response of the Southern Atlantic Anticyclone in figure 1. Maybe if you show a broader latitudinal band (at least -40 to 40) or the streamfunction anomalies, the Gill type response will be clearer.

We would like to point out that the Gill-type response in the Southern Atlantic Ocean, which is forced by positive rainfall anomalies to the east in the western tropical Indian Ocean, is cyclonic wind anomalies in the lower-level (clockwise in southern hemisphere), rather than anti-cyclonic anomalies. As shown in Fig. 1e, the cyclonic wind anomalies comprise of strong westerly anomalies in the tropical Atlantic, northerly anomalies off the west coast of South Africa, weak easterly over subtropical south Atlantic at $\sim 15^{\circ}\text{S}$, and southerly anomalies and South America. Since 40°N or 40°S are far away from the equator and exceed the Rossby radius of deformation, we did not expand the latitude band in the figure.

We also would like to note that as part of the anomalous cyclone, the tropical westerly wind anomalies are particularly strong compared to other components. This is because in low latitudes, the Coriolis force is negligible and as a result the pressure gradient force can more effectively drive wind changes compared to mid and high latitude regions. Indeed, the classic schematic diagrams from Gill (1982) and many other studies also show that the tropical wind response as part of the Gill-type response is much stronger than off-equatorial regions, which is consistent with the results shown in Fig. 1. We have made clarifications in the revised manuscript (L81-86).

Figure S4: I can't see clear propagation of the Kelvin waves in any of the panels. Moreover, 1995 and 1998 are positive Atlantic Niños and have more or less opposite patterns of sea level anomalies in figure S4. Could you please mark the waves in the figure and explain better the differences?

In response to the comment, we have added arrows to mark the propagation of Kelvin waves. Note that since we now use the new definition for Atlantic Niño, the selected years are different. As shown in Fig. R10b and R10c, the two Atlantic Niño events are associated with positive sea level anomalies propagating from the western basin to the eastern basin in both cases. Similar for the four Atlantic Niña events (Figs. R10d-g). This figure is now added to the revised manuscript.

Lead-lag regression of SST & 850hPa winds on DMI

Figure R1 Lead-lag regression of monthly SST anomalies (SSTA; shading; $^{\circ}\text{C } ^{\circ}\text{C}^{-1}$) from HadISST on monthly DMI for the 1950-2019 period. (a)-(h) show results for -7 to 7 months leading time, with negative representing prior to IOD peak and positive for after IOD peak. Note that ENSO-associated signals have been retained, which is different from the results shown in Figure 1 of the original MS.

Lead-lag regression of SST on SON-DMI

Figure R2 Lead-lag regression of 3-month mean SSTAs on September–November (SON)–mean DMI. Units are $^{\circ}\text{C } ^{\circ}\text{C}^{-1}$. (a)–(f) show results for –4 to 6 months leading time, with negative represent prior to IOD peak. Shown are results that are statistically significant at the 90% confidence level.

Figure R3 Composite differences of 3-month running mean SSTAs between selected Atlantic Niño and Niña events that are associated with the positive and negative IOD forcing, respectively. Unit is °C. (a)-(f) show results from month -5 month 0, with month 0 representing the peak of the Atlantic Niño. Shown are results that are statistically significant at the 90% confidence level.

Figure R4 Evolutions of 3-month running mean DMI during selected (a) pIOD-associated Atlantic Niño and (b) nIOD-associated Atlantic Niña events. Shown are DMI from month -5 to month 0 , with month 0 representing the peak of Atlantic Niño/Niña.

Figure R5 SST (shading; °C) and 850hPa wind (vector: m s^{-1}) anomalies in 2019 in the ensemble mean of Indian Ocean pacemaker experiments. (a)-(e) Results from August to December 2019.

Figure R6 Blue shading represents the nudging region in the Indian Ocean pacemaker experiments where the SSTA is restored to observed values. A sponge layer is added at the western, southern and eastern boundaries.

Figure R7 Peak of 3-month running mean ATL3 indices during Atlantic Niño and Niña years in ensemble mean of Indian Ocean pacemaker experiments. Unit is °C. Blue bars represent Atlantic Niño or Niña that are associated with the IOD forcing, while red bars are for independent Atlantic Niño or Niña events. Numbers of the events for each category are shown at the top or bottom of the bars.

Figure R8 Frequency of (a) Atlantic Niño or (b) Niña events occurring each month in observations, which are shown in Fig. R4. Unit is %. Blue represents the Atlantic Niño/Niña events that are associated with the IOD forcing, and red represents the independent events.

Figure R9 Lead-lag regression of SST (shading; $^{\circ}\text{C}$) and surface wind stress (N m^{-2}) anomalies in the tropical Atlantic Ocean and precipitation anomalies (shading; mm day^{-1}) in the tropical Indian Ocean on monthly DMI. Shown are ensemble mean Indian Ocean Pacemaker experiment results that are statistically significant at the 90% confidence level. (a)-(f) show results for -5 to 5 months leading time, with negative representing DMI lagging and positive for DMI leading.

Figure R10 (a) Regions used to calculate the sea level anomalies in (b)-(g). (b)-(g) Hovmöller diagrams showing sea level anomalies along the Atlantic equator and along eastern boundary, the red and blue regions shown in (a). Unit is cm. The annual mean values have been removed at each grid point. Red and blue arrows mark the propagation of equatorial and coastal Kelvin waves during Atlantic Niño and Niña, respectively.

REVIEWER COMMENTS

Reviewer #1 (Remarks to the Author):

I believe the authors have done an excellent job addressing my comments. Making the definition of the Atlantic Nino independent of season is a good choice. Judging from the lagged regressions on the SON DMI index, the IOD mostly contributes to the DJF Atlantic Nino. This has also been called Atlantic Nino II by Okumura and Xie (2006), and has also been recently termed as late-onset Atlantic Nino by Valles-Casanova et al. (2020). It may be good to specifically mention this fact in the manuscript, if it has not been done already. I have no further comments.

Reviewer: Ingo Richter

Reviewer #2 (Remarks to the Author):

The authors addressed my comments satisfactorily. This version of the manuscript improved considerably, in particular the extra analysis and the addition of the pacemaker experiments to the manuscript strengthened the results. I have no further comment.

Reviewer #3 (Remarks to the Author):

Second review of NCOMMS-21-06746 "Indian Ocean Dipole leads to Atlantic Niño" by Zhang and Han, submitted to Nature Communications.

The authors have addressed my main concern about the model experiment and I find that the manuscript has improved, the results are novel and deserve publication. Nevertheless, I still think that there are some major points to tackled before publication.

The seasonality of the connection needs to be addressed. The Atlantic Niño tends to peak in summer, but there is a winter second peak as well. From your results it would seem that the IOD would impact in both seasons, although more in the winter peak, which is very interesting, but not really tackled in the manuscript. I think that you should separate the two seasons and discuss if the mechanisms at work are similar in both. Okumura and Xie (2006) showed that the seasonality of the ocean-atmosphere state in the Atlantic is similar in both seasons (increase in easterly winds in the equatorial Atlantic, shoaling of the Atlantic thermocline and strengthened Bjerkens feedback) but, are the climatological conditions in the tropical atmosphere similar in both seasons prone to allow the teleconnection between the Indian and Atlantic in the same way? The regions of maximum convection in the Indian Ocean in boreal summer and winter are not the same, does this affect the Atlantic response to the IOD in each of the seasons?

Regarding the Gill response, I am sorry but I still don't see the Gill response and the Rossby wave in figure 1. I strongly suggest to plot streamfunction and velocity potential anomalies in the top of the troposphere to see it. This can be done for boreal summer and winter, to address the above.

I find difficult to interpret figure S12 in terms of absolute numbers, so I have plotted a similar figure but with the actual number of events for each month. It is clear from that figure that the strongest impact of the IOD is in Atlantic Niños in winter (70% of December events and 100% of

January events). It is also clear that the relative importance of the IOD in the total boreal summer Atlantic Niños/as is small (24,6%) compared to winter (53.4%). Maybe you can comment on this and focus in the winter Atlantic Niño/a events.

How are the SST in the Indian Ocean in the pacemaker experiments? Please add a panel with the Indian Ocean SST in figure 5. You could also show the SSH anomalies for the model in order to see the wave propagation.

References:

Okumura, Y., & Xie, S. P. (2006). Some overlooked features of tropical Atlantic climate leading to a new Niño-like phenomenon. *Journal of climate*, 19(22), 5859-5874.

Response to Reviews

We are glad that the reviewers have found our initial responses overall satisfactory. We also appreciate the additional constructive comments from Reviewer #3, which have helped us to further improve the manuscript. Below, we respond to the reviewers' comments sequentially, repeating each comment in "black". Our responses are in blue.

Reviewer #1

I believe the authors have done an excellent job addressing my comments. Making the definition of the Atlantic Niño independent of season is a good choice. Judging from the lagged regressions on the SON DMI index, the IOD mostly contributes to the DJF Atlantic Niño. This has also been called Atlantic Niño II by Okumura and Xie (2006), and has also been recently termed as late-onset Atlantic Niño by Valles-Casanova et al. (2020). It may be good to specifically mention this fact in the manuscript, if it has not been done already. I have no further comments.

Thank you for the helpful comments. We have now explicitly discussed the effect of IOD on Atlantic Niño II or late-onset Atlantic Niño in the revised manuscript (L191-196).

Reviewer #3

The authors have addressed my main concern about the model experiment and I find that the manuscript has improved, the results are novel and deserve publication. Nevertheless, I still think that there are some major points to tackled before publication.

We are glad that the reviewer has found our results novel and deserve publication, which are encouraging. We also appreciate the additional constructive comments, which have led to further improvement of our manuscript. Below we have provided point-by-point responses, based on which we have carefully revised the manuscript accordingly.

The seasonality of the connection needs to be addressed. The Atlantic Niño tends to peak in summer, but there is a winter second peak as well. From your results it would seem that the IOD would impact in both seasons, although more in the winter peak, which is very interesting, but not really tackled in the manuscript. I think that you should separate the two seasons and discuss if the mechanisms at work are similar in both. Okumura and Xie (2006) showed that the seasonality of the ocean-atmosphere state in the Atlantic is similar in both seasons (increase in easterly winds in the equatorial Atlantic, shoaling of the Atlantic thermocline and strengthened Bjerknes feedback) but, are the climatological conditions in the tropical atmosphere similar in both seasons prone to allow the teleconnection between the Indian and Atlantic in the same way? The regions of maximum convection in the Indian Ocean in boreal summer and winter are not the same, does this affect the Atlantic response to the IOD in each of the seasons?

I find difficult to interpret figure S12 in terms of absolute numbers, so I have plotted a similar figure but with the actual number of events for each month. It is clear from that figure that the strongest impact of the IOD is in Atlantic Niños in winter (70% of December events and 100% of January events). It is also clear that the relative importance of the IOD in the total boreal summer Atlantic Niños/as is small (24.6%) compared to winter (53.4%). Maybe you can comment on this and focus in the winter Atlantic Niño/a events.

Since the two comments are related, here we discuss them together. We appreciate the constructive comment, and totally agree that the IOD seems to mainly affect the Atlantic Niño that peaks during boreal winter (referred to as Atlantic Niño II in Okumura and Xie, 2006). The reason for such seasonality of IOD-Atlantic Niño connection is that the IOD generally develops in boreal summer and peaks in the fall. As a result, the IOD has a stronger effect on the Atlantic Niño II during boreal winter after the IOD has fully developed and reached its peak amplitude, while it has a weaker effect on Atlantic Niño during summer when the IOD has a weaker amplitude during its developing phase.

In addition, the different mean state conditions in the tropical Indian Ocean between summer and winter may be another reason, as pointed out by the reviewer. In particular, Fig. R1 shows that SST in the western tropical Indian Ocean is lower during summer under the influence of Indian summer monsoon, but higher during fall and winter seasons. Given the nonlinear dependence of rainfall on SST, the higher background SST allows the tropical Indian Ocean SSTA to more effectively affect local rainfall, which then subsequently affects the Atlantic Niño through the atmospheric teleconnection. We have now explicitly mentioned that the IOD may more effectively affect Atlantic Niño II, and have added the discussions above in the revised manuscript (L191-196 & 201-207). We have also modified Fig. S12 to show the actual number of Atlantic Niño and Niña events that are associated with and are independent of the IOD forcing. Thank you for the constructive comments, which certainly have helped improve the manuscript.

Regarding the Gill response, I am sorry but I still don't see the Gill response and the Rossby wave in figure 1. I strongly suggest to plot stream function and velocity potential anomalies in the top of the troposphere to see it. This can be done for boreal summer and winter, to address the above.

In response to the comment, we have shown the 200hPa stream function calculated using lead-lag regression of 200hPa wind anomalies on DMI (Fig. R2). There is a pair of anti-cyclonic anomalies over the western tropical Indian Ocean and tropical Atlantic Oceans from ~10E-60E, between which there are prominent upper-level easterly wind anomalies. The anomalous anti-cyclones are mainly located at the western tropical Indian Ocean in month -5 (Fig. R2a), which then extend to the tropical Atlantic Ocean and meanwhile strengthen (Fig. R2b-e). Please note that ENSO influences have been removed prior to the calculation of regression of wind anomalies on the DMI, since ENSO can also significantly affect the upper-level winds over the tropical Atlantic Ocean. These wind anomalies are clear Rossby wave responses to the warm SSTA in the western tropical Indian Ocean, and we hope this new figure will address the reviewer's concern.

How are the SST in the Indian Ocean in the pacemaker experiments? Please add a panel with the

Indian Ocean SST in figure 5. You could also show the SSH anomalies for the model in order to see the wave propagation.

We have now shown the regression of SSTAs in the pacemaker experiments (Fig. R3), and added to the manuscript as a supplementary figure (Fig. S11 of the revised manuscript). As expected, the evolution of tropical Indian Ocean SSTAs in the model agrees with rainfall anomalies (Fig. 5) very well. In addition, since the oceanic wave propagation associated with the Atlantic Niño is quite fast (propagate through the entire basin in ~1-2 months), it cannot be captured by the regression analysis of monthly SSH anomalies. Hence, we have not shown the SSH anomalies in the model. We hope the reviewer would agree with our decision.

Figure R1 Seasonal climatology of SST (shading, °C), 850hPa winds (vector, m s⁻¹), and precipitation (contour, mm day⁻¹). Contour interval is 2.5 mm day⁻¹. (a)-(d) Average during March-May (MAM), June-August (JJA), September-November (SON), and December-February (DJF).

Lead-lag regression of SF200 on DMI

Figure R2 Monthly 200hPa stream function anomalies (shading, $10^5 \text{ m}^2 \text{ s}^{-1} \text{ } ^\circ\text{C}^{-1}$) calculated using lead-lag regression of 200hPa wind anomalies (vector, m s^{-1}) from ERA-20C on monthly DMI. ENSO influences have been removed prior to the analysis by removing the 3-month lead-lag regression on the Niño-3.4 index. (a)-(h) show results for -5 to 5 months leading time, with negative representing months prior to IOD peak and positive for after IOD peak.

Figure R3 Lead-lag regression of SST anomalies (shading; °C) on monthly DMI. Shown are ensemble mean Indian Ocean Pacemaker experiment results that are statistically significant at the 90% confidence level. (a)-(f) show results for -5 to 5 months leading time, with negative representing months prior to IOD peak and positive for after IOD peak.

REVIEWERS' COMMENTS

Reviewer #3 (Remarks to the Author):

The authors have addressed all my concerns, so I think the manuscript is now ready for publication.

Reviewer #3

The authors have addressed all my concerns, so I think the manuscript is now ready for publication.

We are glad that the reviewer has found our responses satisfactory. We also appreciate the helpful comments from previous rounds that have helped improve the manuscript.